# Generalisation Gap of Keyword Spotters in a Cross-Speaker Low-Resource Scenario

**DOI:** 10.3390/s21248313

**Published:** 2021-12-12

**Authors:** Łukasz Lepak, Kacper Radzikowski, Robert Nowak, Karol J. Piczak

**Affiliations:** 1Institute of Computer Science, Faculty of Electronics and Information Technology, Warsaw University of Technology, 00-665 Warsaw, Poland; kradziko@fuji.waseda.jp (K.R.); robert.nowak@pw.edu.pl (R.N.); 2Graduate School of Information, Production and Systems, Waseda University, Tokyo 808-0135, Japan; 3Institute of Computer Science and Computational Mathematics, Jagiellonian University, 30-348 Krakow, Poland

**Keywords:** keyword spotting, speech embedding, siamese networks, automatic speech recognition

## Abstract

Models for keyword spotting in continuous recordings can significantly improve the experience of navigating vast libraries of audio recordings. In this paper, we describe the development of such a keyword spotting system detecting regions of interest in Polish call centre conversations. Unfortunately, in spite of recent advancements in automatic speech recognition systems, human-level transcription accuracy reported on English benchmarks does not reflect the performance achievable in low-resource languages, such as Polish. Therefore, in this work, we shift our focus from complete speech-to-text conversion to acoustic similarity matching in the hope of reducing the demand for data annotation. As our primary approach, we evaluate Siamese and prototypical neural networks trained on several datasets of English and Polish recordings. While we obtain usable results in English, our models’ performance remains unsatisfactory when applied to Polish speech, both after mono- and cross-lingual training. This performance gap shows that generalisation with limited training resources is a significant obstacle for actual deployments in low-resource languages. As a potential countermeasure, we implement a detector using audio embeddings generated with a generic pre-trained model provided by Google. It has a much more favourable profile when applied in a cross-lingual setup to detect Polish audio patterns. Nevertheless, despite these promising results, its performance on out-of-distribution data are still far from stellar. It would indicate that, in spite of the richness of internal representations created by more generic models, such speech embeddings are not entirely malleable to cross-language transfer.

## 1. Introduction

### 1.1. Keyword Spotting

The goal of a keyword spotter is to detect words of interest in continuous audio recordings. These recordings can be provided either as prerecorded files of considerable length (offline processing) or real-time streaming data (online processing). Nowadays, keyword spotting is more often associated with the latter scenario. The purpose of such a real-time detector is to catch utterances of a specific wake word provided by the user and activate a fully functional conversation with a personal digital assistant.

However, offline usage of keyword spotting can be of great help when navigating vast libraries of audio recordings, pinpointing short regions of interest without the need to listen through the entire conversation. This feature is especially applicable to the mundane task of reviewing lengthy call centre recordings.

### 1.2. Paper Overview

In this paper, we describe a case study of developing such a proof-of-concept solution for spotting keywords in call centre recordings of a Polish bank. The system’s goal was to provide rapid localisation of predefined words of interest when reviewing call centre conversations with the client, which would help bank employees process clients’ complaints more efficiently.

### 1.3. Contributions

Our main contributions in this work can be summarised as follows:We create a number of experimental protocols (mono- and cross-lingual) for keyword spotting in continuous audio recordings.We prepare additional Polish datasets for training and evaluation purposes. The recordings sourced from YouTube clips are available on request.We evaluate similarity ranking models (Siamese and prototypical networks) in practical keyword detection tasks.We report the results on using the Google speech embedder on Polish data.We compare the embedding spaces generated by different combinations of the models and training data.We highlight the gap between popular keyword classification benchmarks and performance in more practical tasks (analysing authentic call centre recordings).

### 1.4. Approaches to Keyword Spotting

In general, there are two main approaches that we can utilise for creating keyword spotters: speech-to-text conversion or audio similarity matching. Both of them have their strengths and weaknesses.

#### 1.4.1. Speech-to-Text Conversion

The first approach consists of a full-scale automatic speech recognition (ASR) system transforming incoming audio streams into a textual representation. In this way, searching for regions of interest is converted to a trivial problem of finding words in a text stream. This approach has several advantages.

First of all, the generated output has a much more desirable form. Instead of only locating particular keywords, the system provides the user with a complete transcription of the reviewed conversation. Such an output form can then be utilised for numerous additional purposes.

A textual representation also helps when extending the dictionary of the system. As long as the speech recogniser generates a high-quality transcription, localising completely new keywords is trivial and can be performed ex-post.

Typical ASR systems have another advantage over audio content similarity matching, as they combine acoustic and language modelling information. Processing whole sentences enables the model to refine its predictions and eliminate implausible responses based on the probability estimate of the generated word sequence. Language modelling can thus help lower the word error rate (WER) in comparison to the processing of words in isolation.

Unfortunately, the ASR-based approach has several drawbacks. For instance, any errors introduced during the transcription phase are final and cannot be fixed in post hoc analysis by adjusting the detection threshold. Another considerable drawback of training a complete ASR system is the data intensity of this process. Various estimates place the requirements on training data availability at 2000 h of transcribed recordings for a viable system [1] and as much as 10,000 h for production-quality results [2].

While the high cost of model training on such a scale [2] effectively prohibits any further experimentation with the baseline systems, it is not the most problematic factor. Unfortunately, in mid-2021, there were no publicly available resources of transcribed Polish speech that would remotely meet the required scope of such a project. Consequently, using a complete ASR system is still not a viable option when processing Polish speech, despite the numerous advantages of this approach.

#### 1.4.2. Audio Similarity Matching

Due to the aforementioned problems with data availability, we have opted for the second option, i.e., developing a keyword detection system based on the perceptual similarity of acoustic fragments. Figure 1 presents a schematic depiction of this approach. The main principle of such a system lies in sequentially processing small fragments of an input recording. Each extracted fragment is then compared to a set of predefined exemplary patterns representing specific keywords. If a significant similarity is found, the system reports detection of a given keyword at a particular point in time.

As our similarity detector, we employ a Siamese neural network model. It is a metric-learning model that has been widely used in similarity ranking problems and is specifically tailored to problems where examples of new classes are scarce, i.e., to few-shot learning [3]. We also compare Siamese models with prototypical networks [4], another solution popular in the domain of few-shot learning.

The main advantage of such similarity-based approaches lies in more moderate data requirements, both in terms of quantity and quality of annotation. Instead of using detailed transcripts that are costly to procure, we can train detectors by providing pairs of word utterances with a binary “match”/“no match” label, a much more convenient data acquisition setup.

Additionally, focusing on acoustic similarity allows for recognition of proper names that language modelling might filter out due to their rareness. The patterns might even represent data other than voice, as shown by Wang et al. [5]. By computing similarity in the domain much closer to the raw data representation, it would be also possible to search for fragments based on information usually lost during conversion, such as emotion, the rhythm of speech, and tone of voice. However, in this work, we focus solely on phonetic matching of keywords.

### 1.5. Research Hypothesis

Our research hypothesis is that keyword spotting models that employ acoustic similarity matching should allow for keyword retrieval from call centre conversations in a low-resource language setting.

### 1.6. Research Outcome

Our experiments show that, with a relatively diverse pool of examples used to train the acoustic embedding model, the performance of acoustic similarity matching is satisfactory when evaluated on synthetic benchmarks. Unfortunately, while such training resources are obtainable for the English language, we were unsuccessful in creating an equivalently robust model for Polish speech. This performance gap underlines how problematic it is to create keyword spotting systems for low-resource languages.

However, as the pattern matching in such systems relies solely on acoustic similarity metrics, it should be, at least partially, language-agnostic—as evidenced by the results of Wang et al. [5]. Therefore, a plausible solution would be to train a model on a high-resource language and use it directly on the low-resource language, hoping that such perceptual matching will be sufficiently accurate, despite the language difference. To this end, we perform several experiments by initially training models on various standard English datasets and using Polish patterns in further detection steps. Unfortunately, such models exhibit weak generalisation when switching between languages, even when augmented with additional Polish data during training. This problem could indicate that the internal representation created by these models is highly specialised and attuned to the particular dataset.

As a way to potentially overcome this issue, we also evaluate an approach that utilises more generic speech embeddings created with a model provided by Google [6] that was pre-trained on 200 million English audio clips from YouTube. Our analysis shows that this embedder creates a more nuanced internal representation of the keyword classes, capturing additional variability factors present in the dataset. It is also better at detecting Polish patterns, despite being trained on non-Polish speech. Although the handicap of an enormous training dataset definitely helps in the few-shot cross-language setup, the results are still far from stellar. A final evaluation on out-of-distribution target recordings shows viability only for a very limited subset of potential Polish keywords.

### 1.7. Related Works

#### 1.7.1. Keyword Spotting

Since the advent of deep learning models in speech processing, researchers have proposed numerous variants of such approaches for keyword spotting. Many of these models have been smaller-footprint versions of typical video and audio processing solutions. Unfortunately, most initial evaluations of models employing convolutional neural networks [7] and recurrent neural networks [8] used proprietary recordings for training and validation. More recently, by introducing the Speech Commands dataset [9], Warden established a popular public framework for benchmarking keyword spotting systems.

The most prevalent evaluation protocol for Speech Commands v1 employs an 80:10:10 split between training, validation, and testing data, selecting 10 of the 30 words in the dataset as target commands. The remaining 20 words create a single negative category (“unknown word”), while additional ambient recordings serve as a silent class, resulting in 12 classification options. Standard training data augmentation consists of random shifts and noise injection.

The baseline method provided by the author [9] attains a top-1 classification accuracy of 85.4% with a simple convolutional architecture [7] operating on 40-dimensional log-mel filter bank features. Slightly better performance is achievable with convolutional neural networks using 40-dimensional MFCC features [10]. By combining a broad hyperparameter sweep with depthwise separable convolutions, Zhang et al. were able to bring the accuracy up to 95% even with resource-constrained architectures [11].

In a similar vein, de Andrade et al. [12] applied a recurrent neural attention model to the task of speech command recognition achieving an accuracy of 94.1% on Speech Commands v1 with 80-band mel spectrograms. Zeng and Xiao [13] reported similar levels for baseline LSTM/CNN architectures, which they were able to improve up to 96.2% through the application of BiLSTM layers to a DenseNet model. What is even more noteworthy, these kinds of results are attainable even with low-footprint models suitable for speech recognition at the edge with less than 100,000 parameters (96.3% with EdgeSpeechNets [14], 96.1% reported for CNNs with temporal convolution over spectrograms [15], 96.4% with parametrized sinc-convolutions [16], and 97.5% with a bigger version of MatchboxNet [17]). When the number of parameters is not an issue, current state-of-the-art approaches can achieve even higher accuracy levels with MoEx (*moment exchange*) feature regularisation on wide ResNet models (98.0% with WRN-28-10 consisting of 36.5 million parameters) [18].

Considering these stellar results, it might seem that keyword spotting is a solved problem. However, the accuracy of classifying isolated excerpts from the Speech Commands dataset might not translate to actual performance when working on continuous audio streams. Real scenarios involve unknown word boundaries, more variance in noisy conditions, and a highly imbalanced class distribution between trigger words and background information. This fact was brought up even in the original Speech Commands paper [9]. It has shown that a baseline model, with 88.2% accuracy of classifying isolated words from version 2 of the dataset, detects keywords with a 46.0% true positive rate when applied to a synthetically generated one-hour audio stream. While some papers were able to report much more promising results in this area of streaming keyword spotting, they were either limited to a detection of a single wake-word [19] or used proprietary datasets of unprecedented scope, e.g., 1 million training utterances of “OK/Hey Google” [20,21] and recordings of several hundred thousand subjects interacting with Alexa devices [22].

#### 1.7.2. Speech Recognition in Low-Resource Settings

Over the years, many successful approaches to English ASR [2,23,24,25,26,27,28,29] have been proposed in the form of various deep learning models. Unfortunately, these models were trained on datasets containing hundreds or even thousands of hours of transcribed English speech. Simple adoption of these techniques to Polish ASR systems is thus impossible.

There were some previous attempts to create speech recognition systems specifically for the Polish language. However, they are mostly several years old and based on non-neural network methods. Some examples include systems based on hidden Markov models [30], k-nearest neighbours [31], and speech n-grams [32]. Results achieved with these methods are not satisfactory; therefore, they were not considered further in our project.

A recent development in global ASR research is the introduction of models trained on large amounts of unlabelled data. These methods swap the effort in data collection for computational time. By analysing vast collections of raw recordings, they can create more robust and powerful speech representations that can also facilitate knowledge transfer to low-resource languages.

An example of this approach is the wav2vec 2.0 framework introduced by Facebook [33]. It uses a contrastive self-supervised learning procedure to create latent speech representations. After pre-training on unlabelled data, the connectionist temporal classification (CTC) loss is used to refine the model further on downstream tasks with labelled data. The main advantage of this approach is the tremendous drop in the requirements for data annotation. Wav2vec 2.0 can achieve a WER of 2.9% on LibriSpeech *test-clean* with as little as one hour of labelled data. For such an outcome to occur, a high price has to be paid when training the model. The most performant variant consisting of 24 transformer blocks needs the equivalent of roughly two GPU years of training on a V100 to process the 53,200 h of LibriVox recordings.

An extension of this concept has led to the introduction of *XLSR-53* [34], a large model pre-trained on 56,000 h of recordings in 53 languages. It has shown remarkable reductions in phone error rates when evaluated on various languages of the Common Voice dataset, with as little as one hour of labelled data required for fine-tuning the final model. Based on these results, XLSR-53 seems to be the most promising approach for speech recognition in low-resource languages to date. Due to its complexity, recreating such a model requires a considerable implementation effort and access to a GPU cluster. Unfortunately, Facebook released the pre-trained model only near the very end of 2020, after the conclusion of our project’s development. Nevertheless, for future endeavours in Polish ASR, it is an interesting possibility to explore.

However, in our work, we have opted for the help of a different pre-trained solution. In 2020, a team from Google Research released a speech embedding model trained on 200 million 2-s audio clips from YouTube [6]. This convolutional model aims to convert an audio stream into a stream of 96-dimensional feature vectors. Each generated embedding vector encodes speech content in windows of approximately 800 ms, every 80 ms. To ensure the reusability of these embeddings, the model was pre-trained on an arbitrary set of 5000 keywords split randomly over 125 keyword spotters sharing the embedder backbone. After 40 GPU days of training, the embedding backbone has been released on TensorFlow Hub for reuse. The authors have shown that these generic embeddings could limit the amount of data needed for creating a robust Speech Commands keyword classifier. They were also successful in partially substituting real training data with synthesised speech. Therefore, we have chosen this approach as a potential solution for our data availability problem.

#### 1.7.3. Voice Datasets

Various voice datasets suitable for speech recognition and keyword spotting tasks are used for training neural network models. While many datasets are strictly proprietary, a handful of them are distributed with more permissive licenses.

One of the most popular in this area is the already mentioned Speech Commands by Google [9]. Its second version consists of 35 different words in English, spoken by 2618 different speakers, totalling 105,829 utterances. The Speech Commands dataset is available for download from the TensorFlow Datasets catalogue.

Common Voice [35], an initiative of the Mozilla Foundation, is on the way to become the most significant publicly available dataset of voice recordings. It contains various sentences read by volunteers in many languages and is freely distributed under the Creative Commons Zero license. Everyone can contribute to the development of this dataset by recording their voice through a browser or a mobile phone. The quantity of sentences varies between languages, with English having the most significant share.

Initially, the dataset incorporated new languages when a sufficient number of recordings was amassed. Hence, the Polish version was publicly distributed for the first time only in Common Voice Corpus 5.1, in the later stages of our project’s timeline. Nevertheless, the total length of validated recordings, reaching approximately 100 h, would be insufficient for training a complete ASR system either way. In mid-2021, more than 70 languages are publicly available. However, the distribution of recordings between languages remains very uneven.

Common Voice Single Word Target Segment was a spin-off dataset published alongside Corpus 5.1, resembling an extension of the Speech Commands concept across multiple languages. It contains recordings of spoken digits, as well as the words *yes*, *no*, *hey* and *Firefox*. The number of speakers varies greatly by language.

Another popular voice dataset is LibriSpeech [36]. It is a collection of around 1000 h of sentences in English. Recordings are annotated on a sentence rather than word level, making it more suitable for speech recognition tasks than keyword spotting.

The Spoken Wikipedia Corpus [37] is a collection of Wikipedia articles read by volunteers. It contains 182 h of word-aligned transcriptions in English, 249 h in German, and 79 h in Dutch. Based on this dataset, Wang et al. [5] have successfully employed prototypical networks for few-shot keyword detection, also in a cross-language evaluation. However, their problem formulation assumed that, in the detection phase, the keyword spotter is provided with exemplary patterns belonging to the same speaker as the speaker in the analysed recording. This availability of very similar patterns is in contrast to our project’s assumptions. We expect keyword searching to function equally well in a cross-speaker regime, generalising to completely new speakers not encountered in the training set.

When deploying speech recognition models, some problematic aspects concerning voice datasets have to be considered. For instance, although the datasets described in this section use permissive licenses, many other datasets may come with limits on their research and commercial usage.

Another very significant problem is language availability. Numerous datasets are available in English. However, finding a good dataset in other languages is often not easy, as they tend to have much less data or insufficient quality. This deficiency makes speech recognition and keyword spotting tasks outside English very difficult. Although several authors, apart from the Common Voice project [35], have prepared Polish voice corpora, they are either limited in recording length [38,39,40], not easily downloadable for offline use [41,42] or proprietary [43]. A potentially interesting dataset is SNUV (Spelling and NUmbers Voice database) [44] that contains 220 h of Polish speakers reading numbers and spelling words. However, due to the lack of complete word utterances (only spelling), its usability in training keyword spotters might be limited.

## 2. Materials and Methods

During the development of our keyword spotting system, we have conducted several experiments with various training and evaluation datasets, both in English and Polish. All the experiments in keyword detection adhere to the same pattern.

First, we prepare the datasets by preprocessing the recordings into mel spectrograms and dividing the data into three parts, described in Section 2.1:training data—single-word utterances used for training the acoustic similarity model,search patterns—keyword examples used as templates for comparison with fragments of the analysed evaluation recording,evaluation recordings—longer recordings used to assess how well the model detects keywords.

Following this step, we train the acoustic similarity model on single-word utterances. In our work, we consider two different types of models that we train from scratch: a Siamese convolutional neural network (*Siamese*) and a prototypical network (*Prototypical*). We also use a pre-trained speech embedder (*Google*) for comparison. Section 2.2 describes in more detail the architectures of these models and the training procedure.

After training, the similarity models work by calculating distances between a provided recording fragment and predefined keyword patterns. For each time step, this mode of operation provides distinct similarity values for each of the analysed keywords.

Therefore, as the last step, we use a detector to aggregate all this information into actual detection decisions based on the configured similarity threshold level. In this phase, we also introduce some temporal smoothing and filtering to make the predictions more robust. We provide more detailed detector configurations in Section 2.3.

### 2.1. Datasets and Data Preprocessing

The main goal of our system was to detect 22 pre-selected keywords in Polish call centre recordings provided by our industrial partner. These recordings came as an unlabelled collection of couple hundred conversations of differing lengths (from minutes to more than an hour) registered with standard call centre equipment at 8 kHz. Unfortunately, privacy concerns with this dataset resulted in stringent local access policies that proved to be problematic in combination with the ongoing pandemic restrictions. Therefore, we could effectively use this dataset only to evaluate the final system, with potentially significant domain shift.

In this less than ideal scenario, we had to resort to several other datasets, with different characteristics, for training and interim validation. Some of the datasets contain real audio recordings, while others are generated as synthetic mixtures. The original sampling rates of the datasets are varied, but we initially downsample all data to 8 kHz, which is the sample rate of the target recordings. We provide specific parameters of spectrogram preprocessing for each of the model types in Table 1.

It is worth noting that the Google model has its processing sample rate set to 16 kHz. This setting aims to match the original training configuration [6]. However, as all our source recordings are initially downsampled to 8 kHz, the available information is the same for all models. Moreover, this discrepancy is also negligible as, effectively, the pre-trained speech embedder only uses log-mel features up to 3.8 kHz [6].

#### 2.1.1. Training Data

This section provides short descriptions of the training datasets. We use training folds for creating the similarity model and validation data for interim monitoring. Test folds either serve as the basis for evaluation mixtures or are discarded. We initially downsample all recordings to 8 kHz. Where needed, we also refactor the datasets for the purpose of prototypical training by further splitting the data into *query* and *support* subsets by selecting speakers with a 75:25 ratio. Table 2 presents the aggregated statistics for the datasets.

##### Speech Commands (SC 1_EN_, SC 2_EN_, ΔSC_EN_)

Speech Commands datasets [9] are defined as follows: SC 1 contains words from 30 classes, SC 2 is a superset of SC 1 extended to 35 classes, while ΔSC contains words from the five classes that are present exclusively in SC 2. All recordings contain an utterance of a single word. We follow the original structure of the dataset by using the pre-defined data splits.

##### Common Voice Single Word (CV_EN_, CV_PL_)

Common Voice Single Word datasets [35] are available both in English and Polish. Each recording contains a single word. We prepare the dataset by splitting it into *train*, *validation*, and *test* folds with a 80:10:10 ratio.

##### Spoken Wikipedia Corpus (SWC_EN_)

Spoken Wikipedia Corpus [37] contains English Wikipedia articles read by volunteers. Based on the provided annotations, we extract single word fragments that fit our constraints: they have a sample duration between 200 and 1000 ms, a keyword length of at least three characters, at least five unique speakers for a given keyword, and at least ten samples in total. We limit the number of samples per single keyword/speaker combination to five occurrences. We then split the created collection into 80:20 training and validation folds.

##### Warsaw University of Technology (WUT_PL_)

The Warsaw University of Technology dataset is a small collection of single word utterances based on our list of target Polish keywords extended with 14 classes with high acoustic similarity. The word examples are recorded with personal devices by employees of the Artificial Intelligence Division at the Warsaw University of Technology and the personnel of mBank SA. The dataset is used exclusively for training purposes.

##### Text-to-Speech (TTS_PL_)

The Text-to-Speech keywords dataset contains Polish spoken words generated with Azure and Google text-to-speech services. We use two forms of this dataset. For pattern matching, we limit the keywords to the 22 classes of the target evaluation. For training purposes, we extend the vocabulary with additional words found while scraping the bank’s website. This process results in a total of 5687 classes generated with seven TTS voices.

##### Combined Datasets (ALL_EN_, ALL_PL_)

Combined datasets, as the name suggests, are composed from the datasets described earlier. For the English language, we merge Speech Commands v2 (SC 2_EN_), English Common Voice Single Word (CV_EN_) and Spoken Wikipedia Corpus (SWC_EN_). For the Polish language, the combined dataset consists of the Polish Common Voice Single Word (CV_PL_), Warsaw University of Technology (WUT_PL_) and Text-to-Speech recordings (TTS_PL_).

#### 2.1.2. Search Patterns

Herein, we briefly describe the collections of recordings that we use in our experiments as keyword templates in the matching process. For each keyword, we use 10 recordings as templates.

##### Speech Commands (SC 1_EN_, ΔSC_EN_)

Speech Commands search patterns are generated directly from the respective SC 1 and ΔSC datasets. For each keyword, we select ten random examples from the datasets’ merged training and validation folds. The datasets consist of 30 and 5 classes, respectively.

##### Target Keywords (KW_PL_)

We create these search patterns based on the Warsaw University of Technology and Text-to-Speech recordings. The dataset contains ten examples for each of the 22 target keywords, although only 19 keywords are actually present in the evaluation recordings.

#### 2.1.3. Evaluation Recordings

This section describes the datasets we use for evaluation purposes.

##### Synthetic Mixtures (SC 1_EN,mix_, ΔSC_EN,mix_, KW_PL,mix_)

We create synthetic evaluation recordings by combining the utterances from the Speech Commands datasets (SC 1_EN_ and ΔSC_EN_) as a single continuous audio stream. The utterances come from the respective test folds. In each case, we place 20 keyword samples per recording with random delays between each occurrence. We generate 50 random evaluation mixtures in this way. For the Polish language, we generate these mixtures based on 1136 hand-annotated keyword examples from YouTube audio clips. All these synthetic mixtures contain only keywords detected in a given scenario, and are merged into a continuous audio stream by us.

##### Speech Commands Overlay on VoxCeleb (Semi-Synthetic) (SC 1_EN,Vox_)

This evaluation dataset consists of Speech Commands keywords overlaid on various backgrounds in the form of VoxCeleb [45] conversations. For each evaluation recording, we use a single keyword occurrence and two conversation fragments with a total length of approximately 15 s. We generate 20 recordings for each keyword.

##### Target Keywords in YouTube Recordings (Authentic) (KW_PL,real_)

As our most realistic evaluation protocol, we use authentic fragments of continuous speech extracted from various YouTube audio clips. Each fragment contains at least one target keyword and lasts from a couple of seconds to more than a minute. We annotated these fragments by hand. The recordings are separate from the synthetic mixtures (KW_PL,mix_). These recordings are real and contain numerous background words which are not the keywords we wish to detect. In total, we use 344 recordings with 511 keyword occurrences.

##### Call Centre Target Keywords (CC_PL,real_)

This dataset was provided by our industrial partner. It consists of several hundred recordings from the bank’s call centre, with conversation lengths varying from seconds to hours, sparsely distributed occurrences of keywords and noise typical for call centre recordings. The recordings are provided in a typical call centre quality, an 8 kHz sample rate, and are unlabelled. During the project’s time frame, the keyword labelling was performed for about 20% of the provided recordings. The availability of the dataset was strictly limited due to stringent privacy regulations, so we only tested our final approaches on it, as it required much coordination and effort to launch it on the bank’s infrastructure.

### 2.2. Similarity Ranking Models

In this part, we briefly provide the specifications for our model and training procedures.

#### 2.2.1. Siamese Convolutional Neural Network

As our primary similarity ranking model, we use a Siamese neural network. The goal of this model is to map pairs of examples into pairs of embedding vectors. During optimisation, embeddings of examples belonging to the same class are placed close together in the embedding space.

The network consists of two identical convolutional branches serving as a speech embedder. In practice, we employ only one instantiation of the convolutional embedder as both branches are identical clones, with shared parameters. The convolutional embedder uses a VGG-like architecture, depicted in Figure 2, with four convolutional blocks, two dense layers, and a linear embedding. Each block combines 3×3 padded convolutions with batch normalisation, Leaky ReLU activations, max pooling, and 10% dropout. We use 32–64–128–64 filters, accordingly. After each dense layer (128 and 256 neurons), we add batch normalisation, Leaky ReLU, and 10% dropout. The final embedding has an output size of 128 features. We normalise the input recordings with 10% dropout and single-channel batch normalisation.

The Siamese model training is based on the contrastive loss, i.e., for pairs of training examples $\langle X_{i},X_{j}\rangle$ created with a given selection strategy ($\langle i,j\rangle \in P$), we calculate the loss value according to the following formula:(1)$$L\left(W\;,\;X,Y\right)=\frac{1}{\left|P\right|}\sum_{\langle i,j\rangle \in P}\left(\left(1-Y_{\langle i,j\rangle }\right)\cdot D_{W}\left(X_{i},X_{j}\right)+Y_{\langle i,j\rangle }\cdot \max{\left(0,\;m-\sqrt{D_{W}\left(X_{i},X_{j}\right)}\right)}^{2}\right)$$
where W specifies the weights of the embedder, $Y_{\langle i,j\rangle }$ term defines if the examples ($X_{i},X_{j}$) are similar ($Y_{\langle i,j\rangle }=0$) or dissimilar ($Y_{\langle i,j\rangle }=1$), $D_{W}$ is the similarity (distance) function defined for the embeddings $\langle G_{W}\left(X_{i}\right),G_{W}\left(X_{j}\right)\rangle$, and *m* is the margin value. In our case, we use squared Euclidean distances, i.e.,:(2)$$D_{W}\left(X_{i},X_{j}\right)={∥G_{W}\left(X_{i}\right)-G_{W}\left(X_{j}\right)∥}_{2}^{2}$$

This loss formulation keeps the embeddings of the same class closer together. On the other hand, the embeddings of different keyword classes are pushed away, so that they do not fall inside the margin, as illustrated in Figure 3.

We train the model for 100 epochs, using the Adam optimiser with default hyperparameters and a learning rate of 0.001, with a contrastive loss margin of 1.0. We use 50,000 training examples per epoch. In each epoch, we generate 200 training episodes, consisting of 25 classes per episode and 10 samples per class. Our pair selection procedure uses the *hard negative* variant, i.e., we first create all possible positive pairs (matching keywords) from the samples in the current episode, and then we generate an equal number of negative pairs selecting examples with the smallest distances.

#### 2.2.2. Prototypical Network

In prototypical network experiments, we use the same embedder architecture as in Section 2.2.1. The main difference between the Siamese and prototypical approaches lies in the introduction of class prototypes. In Siamese models, we calculate the distances for similar/dissimilar pairs of individual examples from the dataset, whereas prototypical networks divide the training data into *support* (S) and *query* (Q) subsets. In each episode of training, a class prototype is created for each of the *C* selected classes by computing the mean value of the *K* vectors from the support subset:(3)$${\bar{S}}_{c}=\frac{1}{K}\sum_{i=1}^{K}S_{c_{i}},$$
where $S_{c}$ denotes support examples of class *c*. The pair comparison is then performed between such class prototypes and all query examples selected in the given episode, resulting in a loss function: (4)$$L\left(W\;,\;S,Q,Y,C\right)=\frac{1}{C|Q|}\sum_{c=1}^{C}\sum_{i=1}^{\left|Q\right|}\left(\left(1-Y_{c_{i}}\right)\cdot D_{W}\left({\bar{S}}_{c},Q_{i}\right)+Y_{c_{i}}\cdot \max{\left(0,\;m-\sqrt{D_{W}\left({\bar{S}}_{c},Q_{i}\right)}\right)}^{2}\right),$$
where $Y_{c_{i}}$ defines if the query example $Q_{i}$ is of the same class *c* as the prototype ${\bar{S}}_{c}$ ($Y_{c_{i}}$ = 0) or different ($Y_{c_{i}}$ = 1). A more intuitive depiction of this procedure is presented in Figure 3.

Using this loss formulation, we train the prototypical model for 200 epochs comprising 100 episodes each, by optimising the prototypical loss with squared Euclidean distances and a margin of 1.0. In each prototypical training episode, we take 5 support samples and 15 query samples per class. Each episode consists of 25 classes. This setup results in the same total number of parameter updates for the Siamese and prototypical approaches.

#### 2.2.3. Google Speech Embedder

In the experiments involving the Google speech embedder, we use the pre-trained model provided as version 1 on the TensorFlow Hub (https://tfhub.dev/google/speech_embedding/1, accessed on 10 December 2021). When fine-tuning the network, we use the same training approach as in Section 2.2.1, but with 50 epochs and 10 classes per episode.

### 2.3. Detector Settings

Using the similarity models described in Section 2.2, we generate distances between consecutive fragments of the analysed recording and each of the provided keyword templates (we use 10 templates per keyword class). This way, we obtain multiple values for each single time step, telling us how closely the current fragment resembles these various patterns. To aggregate such data into more meaningful detection decisions, we use various detection policies that we briefly describe in this section.

#### 2.3.1. Common Detection Pipeline

In most of the experiments, we use the same detection pipeline. We analyse each frame of the recording, i.e., we employ a step size of 1. The actual frame size in milliseconds is determined by the spectrogram processing settings defined in Table 1. We then transform the generated distances with a median filter with a window length of 5 frames, obtaining smoothed similarity scores for each search pattern.

Based on these scores, we find the lowest average across all the analysed classes. If the average score falls below the threshold value defined for a given policy, we emit a detection marker for a given keyword at this particular time step. These marker emissions are then smoothed with a median filter applied across 15 frames.

Finally, if any keyword generates a continuous sequence of markers exceeding our minimum length of 25 consecutive frames, we return a detection at a given time step. We also introduce a minimal distance of 10 frames between two consecutive occurrences of the same keyword.

When evaluating the results of the system, we count the detection as a true positive if it falls at a time step representing the middle of the keyword occurrence, with a collar of 1 s. We adjust the policy threshold values based on the specifics of each model. For instance, for the Siamese model, we evaluate all values from 0 to 1 with a step size of 0.05, as this range allows us to generate a complete precision–recall curve.

#### 2.3.2. Additional Post-Processing for the Google Speech Embedder

In standard experiments with the pre-trained embedder model, we use the same approach as described in Section 2.3.1, albeit with a minimum length of consecutive matches reduced to 10 frames. However, our experiments show that the similarity scores returned by direct distance calculations on speech embeddings generated by the Google model have a different scale and are less uniformly distributed across the different keyword classes.

Therefore, we introduce an additional post-processing step to our detection pipeline in the form of an *exciter* module. The role of this module is to make the similarity values more uniform across different keywords and filter out superfluous detections. To achieve this goal, we perform several operations.

First, we standardise the values of the distances in the detection matrix, visualised as input data in Figure 4. We perform this standardisation per each of the 10 patterns (rows). Then, we filter out all the values above the 5th percentile, leaving only the responses for the time steps with the closest matches. This way, we obtain several time series representing filtered and standardised similarity scores for single patterns.

After that, we use an envelope follower to perform excitation filtering on each of the time series. Therefore, after encountering a distance value below our threshold (signifying a potentially close match for a given template), we sustain this information in time with a decaying impulse response for up to 50 steps. When we find another close match, the decaying response is restarted. The result of this process is depicted in Figure 4.

### 2.4. Statistical Analysis of the Results

After running the experiments, we conduct a statistical analysis of the obtained results. For every experiment, we report training, patterns and evaluation datasets that were used. As the performance metrics, we use the Area Under Precision-Recall Curve (AUPRC) and F-score. The AUPRC is calculated based on the precision and recall values for different detection thresholds. Precision tells us how many of the reported keyword detections are correct, while recall shows us how many of the actual keyword utterances are retrieved. The AUPRC is provided with both micro- and macro-averaging. Micro-averaging takes into account the sizes of every keyword class we wish to detect by averaging over all the examples as a whole. This approach might be better suited to our problem, as we mainly deal with imbalanced datasets. On the other hand, macro-averaging, which aggregates the AUPRC values by first calculating them separately for each class, may also be helpful when interpreting the results. The F-score is defined as $F=\frac{\mathrm{precision}\;+\;\mathrm{recall}}{2}$, and it combines the precision and recall results into one metric, with higher values considered better.

## 3. Results

### 3.1. Isolated Words Classification Benchmark

Before proceeding with the actual keyword detection experiments, we performed a baseline verification to make sure that our embedder architecture (*Siamese*), described in Section 2.2.1, is sufficiently performant when processing audio data. To this end, we recreate the Speech Commands evaluation protocol [9] of classifying single-word utterances as one of the 12 possible classes. This is the only experiment where we do not downsample the input recordings. Instead, we maintain the original 16 kHz sampling rate, adjusting the spectrogram preprocessing settings accordingly.

Our convolutional model achieves a 94.5% top-1 classification accuracy on the Speech Commands v1 dataset. This performance is on par with similar models processing mel spectrograms, as presented in Table 3. While more sophisticated models can achieve better classification results, we deemed the differences not significant enough to warrant the trade-off these models introduce in terms of complexity and training time.

We also verify the same architecture on a downsampled version of the Speech Commands recordings. As expected, this impairs the accuracy of the evaluated model. However, the results show that 8 kHz recordings can still provide sufficient information for proper classification.

### 3.2. Keyword Detection in a Monolingual Setup

As a first step in evaluating keyword detectors on continuous audio data streams, we analyse our Siamese approach in monolingual scenarios, i.e., by training and validating the model on recordings of the same language. For selected experiments, we also compare its performance to a prototypical network.

In each evaluation setting, we create a separate *precision-recall curve* (PRC) for each of the analysed keywords. The PRC shows detection performance at different values of the similarity threshold. We aggregate this information across classes using either micro- (*instance*) or macro- (*class*) averaging, a common approach in multi-class problems [46].

To summarise the performance of a model with a single value, we report the *Area Under Precision–Recall Curve* (AUPRC). We also highlight the best F-score achieved by the model across different threshold values. Table 4 presents the results obtained in the analysed monolingual scenarios.

Our initial verification, described in Section 3.1, has shown that a Siamese convolutional neural network can effectively differentiate between utterances of different keywords of the Speech Commands dataset. We also confirm this capability in a streaming evaluation by employing the Siamese model with standard detection settings on a synthetic mixture of Speech Commands keywords (SC 1_EN,mix_). When the vocabulary present in the target recording consists solely of the expected keywords, the model can achieve an outstanding performance of 91.3% micro-AUPRC and an F-score of 0.94. This result shows that our detection approach allows for properly recognising keywords with shifted word boundaries.

To measure the robustness of the detector to background distractors, we evaluate it on Speech Commands keywords mixed into fragments of conversations of the VoxCeleb dataset (SC 1_EN,Vox_). In this case, the detection accuracy drops to a level of 60.4% micro-AUPRC. Nevertheless, the performance is satisfactory for a cross-speaker keyword spotter as the system can correctly highlight more than 23 bevelledtrue of keyword occurrences while maintaining a precision of 55%. Results at this level would be usable for prospective users of such a tool. Unfortunately, we have to admit that the semi-synthetic nature of this evaluation oversimplifies the task presented to the detector, making it an upper bound on achievable accuracy. Due to the lack of adequate data, we could not verify how such detectors would cope with more natural keyword occurrences and diverse recording conditions.

On the other hand, our problem setting assumes that the detector can only be trained on copious amounts of generic keywords. After that, it should cope well with limited examples of target keywords, especially since the end-user can extend the vocabulary after the system’s deployment. We assess this aspect with an experiment using keywords not occurring in the original training data, isolated from the second version of the Speech Commands dataset (ΔSC_EN,mix_). This evaluation scenario confirms that few-shot learning is quite difficult. A drop of the F-score to 0.65 on synthetic mixtures indicates that, when combined with more natural evaluation settings, detection of completely new keywords might be problematic. A more reasonable approach would require at least some retraining with the extended vocabulary.

Interestingly, across all the experiments, the extension of the Speech Commands training data with other datasets proves detrimental to the model’s performance. We analyse this phenomenon more closely in Figure 5 by visualising the embedding space created with Siamese models trained solely on the Speech Commands data and in combination with the Spoken Wikipedia Corpus recordings. The model trained only on Speech Commands utterances maps the examples from this dataset to groups with much clearer separability between the particular keywords. Apart from some stray confusions, the only intermixing of keywords happens for the “three–tree” pair, showing that the model is indeed focusing on the acoustic similarity of the provided samples. The inclusion of more diverse training examples from the Spoken Wikipedia Corpus prohibits the model from learning an equally discriminative mapping for the Speech Commands keywords. The created groups of examples show much more bleed between keywords. Unfortunately, we were unable to devise a simple mitigation technique for this issue. It is possible that a more nuanced training procedure could create a more robust representation using all of the available data.

We also extend our investigation with additional experiments employing a prototypical network model instead of the Siamese embedder. In the analysed monolingual settings, this approach proves to be less performant than the Siamese counterpart. In Table 4, we report only the results for training with Speech Commands data, but the tendency remains unchanged with different dataset setups.

Finally, we conclude with evaluations performed on Polish datasets. Unfortunately, the experiments confirm our initial concerns. Models trained solely on such limited datasets are entirely unusable in detecting keywords in continuous recordings.

### 3.3. Keyword Detection in a Cross-Lingual Setup

Following the expected failure of models trained exclusively on Polish recordings, we try to approach the problem of detecting Polish keywords with models trained on English datasets, i.e., in a cross-lingual mode. We hope that the general audio processing capabilities acquired by training on more extensive English datasets will allow for a successful transfer of knowledge to Polish recognition tasks, despite the inherent differences between acoustic features of the languages.

Table 5 summarises our findings in cross-lingual scenarios. When trained solely on the Speech Commands recordings, both Siamese and prototypical models are better in detecting Polish keywords than their counterparts trained on limited Polish data. Nevertheless, this improvement is still insufficient to achieve satisfactory performance. The Siamese model, which outperforms the prototypical approach, achieves a micro-AUPRC of only 20.3% on synthetic mixtures of Polish keywords (KW_PL,mix_). In practical terms, this result means that we can roughly achieve a precision of 70% at a 25% recall rate. However, the outcomes for individual keywords vary widely. For instance, the best performing one, *umowa*, has a recall of 67% with 96% precision. Unfortunately, although we can find several other classes with potentially usable results, many keywords have a near-zero detection rate.

When looking at possible training dataset extensions, we observe a similar situation as with monolingual models. Training solely on the Speech Commands dataset proves to be the most efficient way to achieve acoustically discriminative embedders. Additional recordings are similarly detrimental in creating an embedding space appropriate for cross-lingual transfer to Polish patterns.

This observation also holds for further extensions with Polish training recordings, though not without some caveats. Initially, we hoped to fill potential gaps in the generated embedding space by introducing additional training examples more closely resembling the phonetic structure of the target patterns. However, the outcome of this process was somewhat ambiguous.

On the one hand, if we compare the 2D representations of the embedding space presented in Figure 6, the second model, supplied with extended training data, groups Polish keywords into much tighter clusters. Although low-dimensionality mappings of complex embedding spaces might be misleading at times, this visual difference most probably indicates that the second model can more effectively discriminate between the keyword classes.

On the other hand, this capability does not translate to an advantage when comparing the detection performance of both models. While we do not have a definite explanation for this phenomenon, we hypothesise that a more dispersed representation might be actually beneficial in our scenario. In contrast to the evaluation on Speech Commands keywords, where the evaluation and search patterns come from the same distribution, Polish patterns used as keyword templates differ in the recording conditions from the YouTube evaluation fragments. Therefore, a broader, less regularised representation might expose more potential points of contact to find close neighbours that could match actual keyword occurrences in audio streams. Such an increased coverage could be significant since the data manifold of Polish search patterns is quite limited to begin with.

This representational problem is accentuated by evaluations performed on longer, authentic Polish audio streams (KW_PL,real_). In this scenario, all systems fail to provide any hint of usable results. We can devise a two-fold explanation for this behaviour.

First of all, the embedding space learnt by our models does not capture the acoustic differences at a detailed enough level. Therefore, these models cannot handle utterances outside of their limited vocabulary. Non-keyword audio content easily derails the detectors, which is confirmed by numerous false positives.

The second factor is associated with the difficulty of the problem itself. In contrast to our English setups, the evaluation performed on Polish recordings uses longer fragments of naturally sounding speech from diverse recording conditions, thus being the closest to an actual environment in which these kinds of systems might be deployed. Consequently, many keyword occurrences are less perceptible than in semi-synthetic mixtures.

Based on all these observations, we suspect that a self-supervised approach to training acoustic models could be promising in solving similar problems as described in this paper. Self-supervision should help create rich representations, more robustly capturing the differences between various words present in the recording—all without the need for extra hand-labelling. Unfortunately, devising a sensible self-supervised approach is not a trivial task, and each iteration of such an experiment requires a significant computational effort to train the actual model. Therefore, due to the constrained timeline of our project, we were unable to explore this option further.

### 3.4. Keyword Detection with Generic Speech Embeddings

Instead of investigating self-supervised techniques, in this last experimental section, we concentrate on models trained traditionally, in a fully supervised manner, but on much bigger datasets. Based on our assumption that the main factor limiting the performance of our models is the lack of a more generic and robust internal speech representation, we replace our previously analysed similarity models with a pre-trained speech embedding model provided by Google [6]. It has been trained on more than 100,000 h of English audio clips, which we expect should cover a big part of possible recording conditions and variants of speech. We present the results of applying this model in Table 6, divided into four different approaches.

First, we use the pre-trained embedder directly, without any fine-tuning, with the standard detector. This setup means that the pre-trained model returns coordinates in the embedding space for each analysed fragment of the evaluation recording and the keyword patterns. We then compute the distances to each provided keyword example and use properly adjusted thresholds with our primary detection approach involving aggregation and filtering. The macro-results of this system are good on the Speech Commands dataset (SC 1_EN,mix_), although not as good as the specifically trained models. However, the system incorporating the pre-trained embedder is much better with Polish keywords (KW_PL,mix_). It is also the first system to obtain non-zero results on the most interesting, authentic Polish evaluation (KW_PL,real_).

Comparing Figure 5a and Figure 7a, we can observe some significant changes in the embedding space generated for the Speech Commands examples. The Google speech embedder mapping of keyword classes is more complex than in the Siamese model’s case. Instead of densely packed clusters, the pre-trained model distributes the encountered examples across more elongated shapes. This change indicates that the speech embedder captures more variability factors in the data, apart from the keyword class, creating a more nuanced representation. Concurrently, it still creates a distinct separation between the classes, although with some overlap in the middle of the plot. The behaviour in this central region might explain why the raw performance of the pre-trained embedder on Speech Commands data might be slightly worse in direct comparison to a Siamese model.

Despite this robust representation of speech fragments generated by the Google model, a direct application of the returned similarity distances proves to be relatively ineffective in a detection setting. While most keywords produce pretty accurate results, a select few (*eight*, *off*, and *up*) create a tremendous number of false positives. These errors drive down the model’s average performance, indicated by the significant discrepancy between the micro- and macro-values of the reported metrics. This problem is presented more explicitly in Figure A1, in Appendix A.

We mitigate this issue by introducing the post-processing approach described in Section 2.3.2. By adjusting the detector setting in this way, we can almost entirely filter out these massive false detections reported by the model, as shown in Figure A2. This modification comes at a small cost of slightly worse outcomes on some of the high-quality keywords, but on the whole, it creates a detection system with a much more practical behaviour.

The performance of the post-processing version of the system on Speech Commands data (SC 1_EN,mix_) is comparable to our Siamese and prototypical models trained from scratch. The results for semi-synthetic evaluation with SC 1_EN,Vox_ are slightly worse, but the performance on the “*delta*” classes (ΔSC_EN,mix_) is on par with the original Speech Commands keywords. This improvement shows that the Google speech embedder has been exposed to an extensive range of potential English keywords, making it easier to adjust the operating vocabulary on the fly.

The most important change from our perspective is the improvement in Polish keyword detection. A good representation of the Polish keywords, as depicted in Figure 8a, allows the model to approach much more sensible levels even with the baseline detector. However, the post-processing variant brings the best F-score on the KW_PL,mix_ dataset up to 0.69, significantly outperforming all the other approaches analysed in this paper. The macro-AUPRC values on authentic Polish recordings (KW_PL,real_) are better with the standard detector, but post-processing improves both the micro-AUPRC and the F-score value. Unfortunately, it is still somewhat discouraging, achieving a level of 0.17.

Finally, we also evaluate fine-tuned versions of the Google speech embedder, hoping to combine the advantages of both worlds—pre-trained generic representations and dataset-specific mapping. The fine-tuning procedure is described in Section 2.2.3. Unfortunately, our efforts very swiftly prove to be destructible to the intricate representational capabilities of the original model. As evidenced by Figure 7b, fine-tuning on the Speech Commands dataset creates embeddings with some tightly packed clusters, similar to the behaviour of the Siamese model. Still, most of the keywords become intermixed after this procedure. This degradation is also confirmed quantitatively, as the results for the fine-tuned models are comparatively worse across the board. This drop in performance also pertains to fine-tuning on Polish data, although the embedding space shown in Figure 8b seems to be less impacted.

### 3.5. Final Evaluation

As a final step in evaluating our keyword detection systems, we analyse their behaviour on the actual target recordings of call centre conversations (CC_PL,real_). Due to organisational impediments, we could perform this procedure only once, with a limited number of systems. Therefore, we employ the pre-trained embedder solution with our Polish keyword patterns, as such a combination presents the most promising results on continuous Polish recordings. We choose both variants of the system, with the standard detection pipeline and post-processing. The pre-trained embedder is used directly, without fine-tuning.

Table 7 summarises the findings of our final evaluation. The detection pipeline was executed only once with a predetermined detection threshold. Therefore, instead of full AUPRC numbers, we present actual metric values obtained at this sensitivity point.

Both models perform poorly, especially when looking at the micro-aggregation scheme. As the results of our interim validations on Polish datasets were relatively poor, we did not expect a much better outcome, especially since we had to cope with a domain shift on entirely unseen data.

A more detailed analysis shows that the *Google* model returns a very high number of false positives for two short keywords, “*blik*” and “*link*”. As the utterances of these keywords contain only one syllable, it is understandable that matching based only on acoustic features might be unsuccessful. Although our post-processing method can correctly suppress these erroneous detections, it introduces its own biases, keeping the final performance still at a low level.

Concluding our evaluation, we must admit that the quality of predictions generated by the system with the pre-trained embedder proved to be disappointing. This approach was insufficient in creating a general robust keyword spotter for the Polish language. However, there can still be a small added value of such a system when employed as a support tool to highlight specific keywords. Although it will not recall all the occurrences, it can still help people performing the reviewing work if its precision is sufficiently high. We were able to find a couple of characteristic multi-syllable keywords that exhibited promising results in this regard. For instance, words such as “*reklamacja*” and “*transakcja*” had a precision of over 75% combined with a recall rate of 10–20%. While it is not exactly what we have hoped for when devising the system, our proof-of-concept solution has shown that the main focus when creating robust Polish keyword spotting systems should lie on the mundane task of data annotation.

## 4. Discussion

### 4.1. Summary of Findings

In this paper, we have explored the problem of cross-speaker keyword spotting for the low-resource setting of the Polish language.

Our options were limited by the lack of publicly available datasets suitable for training production-quality Polish speech-to-text systems. Therefore, we have focused on spotting keywords with detectors based on acoustic similarity. These approaches are generally less demanding on the data annotation side.

We evaluated two similarity ranking models, i.e., Siamese and prototypical networks. Our experiments with English datasets have shown that these methods can create acoustically discriminative representations of processed recordings when provided with sufficiently diverse training examples. Unfortunately, due to the data scarcity problem, we could not create robust keyword spotters solely on Polish data.

Although the perceptual principles of comparing two audio fragments remain the same on the fundamental level, our acoustic similarity models were unable to generalise from English to Polish. The acoustic differences between languages and recording conditions proved to be too big for such a cross-lingual transfer to succeed.

Therefore, we have evaluated a different approach by utilising a generic speech embedding model provided by Google, extensively trained on thousands of hours of English speech. The advantage provided by a very comprehensive training dataset could be seen in more complex representations of the speech samples and much better adaptability to cross-lingual transfer. Although the evaluation on Polish synthetic recordings was quite promising, even with this pre-trained embedder, we still could not create a system that would be fully functioning in realistic scenarios and could effectively process naturally sounding continuous audio streams.

### 4.2. Future Work

Based on our research findings, we reckon that acoustic similarity comparisons can be a viable approach in various audio matching problems. Nevertheless, the task of creating a robust generic embedding space for speech recordings is not easy, especially when no datasets of considerable size are available for the target domain, as shown by our negative results. This outcome hints at a number of approaches that could be evaluated in future works.

First of all, our evaluations show that simply extending the scope of training data with out-of-domain examples is not always profitable. However, it is possible that we were unsuccessful in finding more effective training methods, better suited for mixed datasets. Techniques, such as domain adaptation of embeddings that have proved successful in NLP tasks [47], could help bridge the gap between models trained on generic datasets and evaluation on target recordings with different characteristics or even across languages.

Looking at the visualisations of the generated embeddings, we see that similarity models can usually maintain correct separation between various classes. Therefore, we expect that, with some careful adjustments to the post-processing schemes, we could improve the quality of the final system. Better ways to discard background noises and erroneous detections in continuous recordings could help utilise the whole potential of the similarity classifiers, which exhibit a good performance in more isolated settings.

However, to achieve these goals, more robust validation procedures and datasets would be needed. This problem is particularly relevant since the disparity between the performance of keyword classifiers and keyword detectors is striking. In fact, during our work, we could not verify the performance of our systems on representative English audio streams. We think that establishing new, more realistic evaluation protocols for keyword spotters would be an interesting extension for future work, and it would be valuable for a broad research community. Our research highlights that the task of searching for individual words in an audio stream is much more challenging to solve than the classification of separated words, which most current methods are benchmarked against.

Additionally, recent developments in self-supervised training of audio representations create exciting opportunities for low-resource languages, such as Polish. We expect that solutions such as XLSR-53 [34] could prove helpful in bridging the generalisation gap that we encounter in low-resource scenarios.

In the end, if no practical improvements can be achieved other than by increasing the sheer amount of data, we think that the introduction of more resource-efficient annotation procedures based on active learning could make such efforts realisable with lower budgets. For instance, one approach that could be employed is the clustering of unlabelled data. The *K*-medoids technique was shown to reduce labelling budgets by half in sound classification tasks [48]. When combined with the feedback of a continuously retrained model, we expect that such solutions could greatly improve the annotation workflow.

## 5. Conclusions

The goal of our work was to create a proof-of-concept solution that could effectively detect Polish keywords in low-quality call centre recordings. Based on our research hypothesis, we developed keyword detectors employing few-shot acoustic similarity models. The models have a satisfactory accuracy in English and for selected Polish keywords, but they fail for many shorter Polish utterances. Effectively, the created software system enables navigation in call centre recordings only for a limited subset of Polish keywords. However, such functionality can still reduce the processing time in the complaint processes.

## Figures and Tables

**Figure 1 sensors-21-08313-f001:**
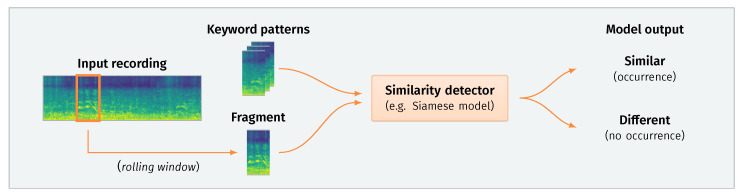
Keyword detection based on audio similarity matching. This schematic depicts a typical pipeline. An input recording is processed sequentially with a short rolling window (e.g., 800 ms). The system compares each extracted fragment to a set of exemplary patterns predefined for each keyword. The model then outputs a decision based on the distance between the fragment and each pattern. If the distance is sufficiently small, a keyword occurrence is emitted for a given timestamp.

**Figure 2 sensors-21-08313-f002:**
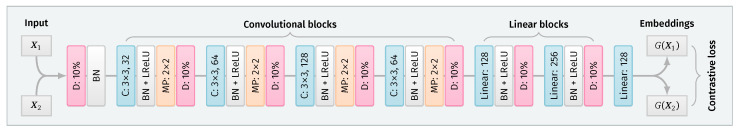
The architecture of the embedding model. Layer types denoted as: D—dropout, BN—batch normalisation, C—convolutional, MP—max-pooling, LReLU—LeakyReLU. Both the Siamese and prototypical models use the same embedder architecture, albeit with different loss functions and input formulation.

**Figure 3 sensors-21-08313-f003:**
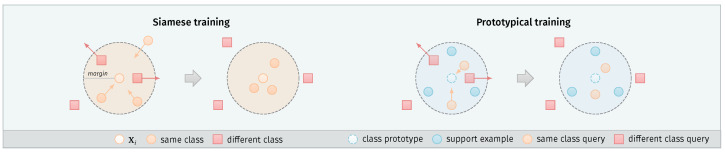
Overview of the Siamese and prototypical training approaches. Siamese networks use a contrastive loss to make pairwise comparisons between examples from the dataset X. Examples from the same class are attracted to each other, while the examples from different classes are pushed back if their distance is lower than the defined margin *m*. The prototypical approach uses additional support examples S. Their mean vector defines the class prototype. Query examples Q of the same class are attracted to this prototype, while queries of different classes are pushed back in the same way as with Siamese training.

**Figure 4 sensors-21-08313-f004:**
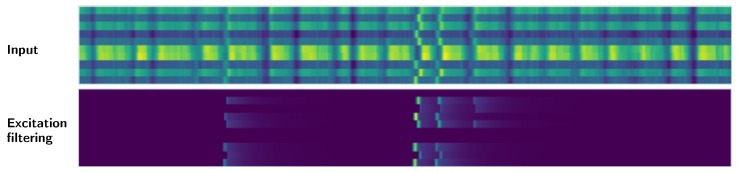
Post-processing of the similarity scores generated by the pre-trained embedder. The first matrix, denoted as *input*, shows an example of raw distances between the embeddings of keyword patterns and recording fragments. Each row corresponds to a single pattern, while each column represents a single time step. Brighter colours indicate smaller distances (closer matches). The second matrix shows the same values after row-wise standardisation and filtering through the *exciter* module.

**Figure 5 sensors-21-08313-f005:**
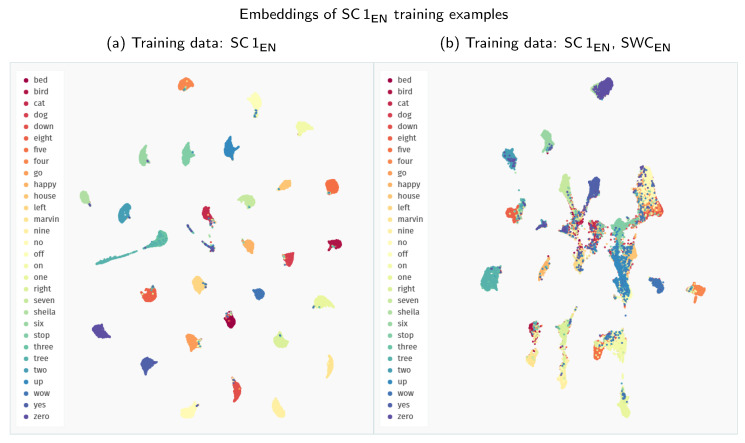
Comparison of the UMAP visualisations of the Speech Commands training examples (SC 1_EN_) processed through the Siamese embedder: (**a**) embeddings generated with a model trained only on the Speech Commands data; (**b**) embeddings generated with a model trained on recordings both from the Speech Commands and the Spoken Wikipedia Corpus datasets. (We employ a zero minimum distance between embedded points. Other visualisation settings use standard values of the *umap-learn* Python package, i.e., 15 neighbours with a Euclidean metric for Uniform Manifold Approximation and Projection.)

**Figure 6 sensors-21-08313-f006:**
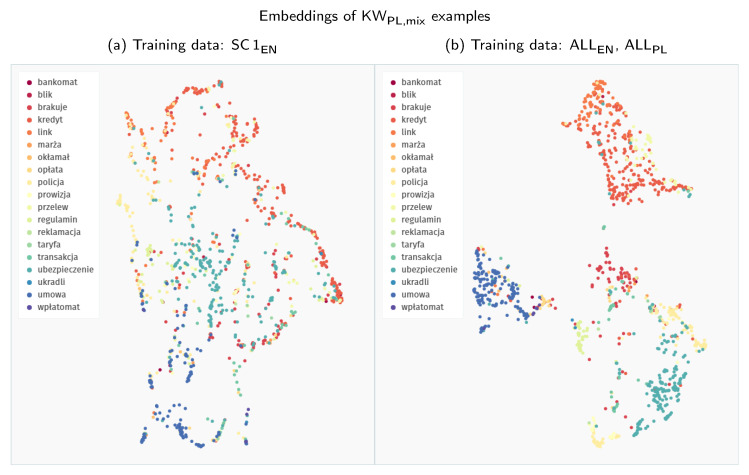
Comparison of the UMAP visualisations of Polish keywords processed through the Siamese embedder: (**a**) embeddings generated with a model trained only on the Speech Commands data; (**b**) embeddings generated with a model trained on all the available data (both English and Polish); Polish keywords are extracted directly from YouTube videos. Synthetic mixtures of these keywords are denoted as KW_PL,mix_ throughout the results section.

**Figure 7 sensors-21-08313-f007:**
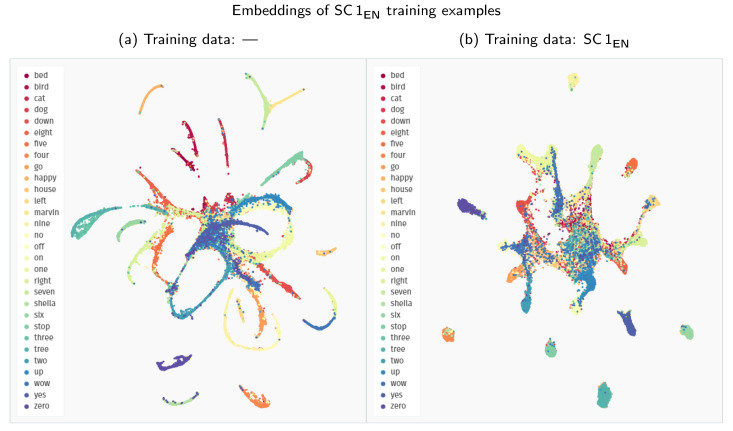
Comparison of the UMAP visualisations of the Speech Commands training examples (SC 1_EN_) processed through the Google speech embedder: (**a**) embeddings generated with a pre-trained model without fine-tuning; (**b**) embeddings generated with a pre-trained model fine-tuned on the Speech Commands dataset.

**Figure 8 sensors-21-08313-f008:**
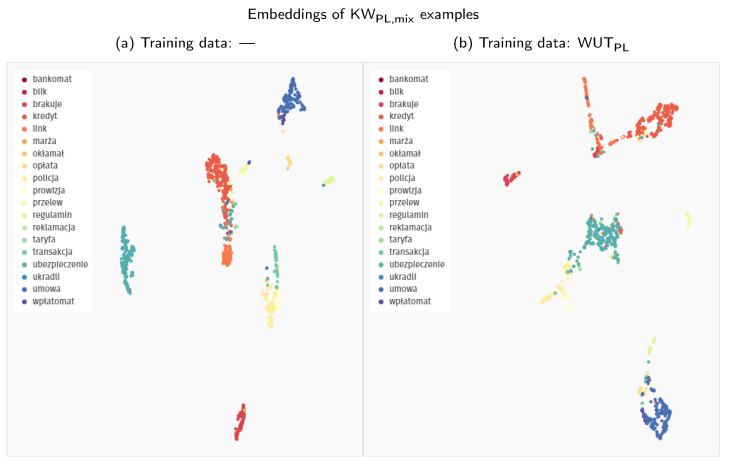
Comparison of the UMAP visualisations of Polish keywords processed through the Google speech embedder: (**a**) embeddings generated with a pre-trained model without fine-tuning; (**b**) embeddings generated with a pre-trained model fine-tuned on WUT_PL_ training examples.

**Table 1 sensors-21-08313-t001:** Parameters of mel spectrogram preprocessing for each model type.

	Siamese	Prototypical	Google
Sample rate	8 kHz	8 kHz	16 kHz
Segment duration	0.8 s	0.8 s	0.775 s
Mel bands	40	40	32
FFT window	512	512	400
Hop length	160	160	160
Centring	false	false	false

**Table 2 sensors-21-08313-t002:** Statistics of the training datasets.

Dataset	Language	Samples	Classes	Speakers	Acronym
Speech Commands v1	English	64,727	30	1881	SC 1_EN_
Speech Commands v2	English	105,829	35	2618	SC 2_EN_
Speech Commands “*delta*”	English	7967	5	663	ΔSC_EN_
Common Voice Single Word	English	26,070	14	3519	CV_EN_
Common Voice Single Word	Polish	898	14	86	CV_PL_
Spoken Wikipedia Corpus	English	429,354	4657	372	SWC_EN_
Warsaw University of Technology	Polish	1058	36	29	WUT_PL_
Text-to-Speech	Polish	39,809	5687	7	TTS_PL_

**Table 3 sensors-21-08313-t003:** Top-1 classification accuracy for isolated words of Speech Commands v1.

Authors		Method		Accuracy
Warden [9]		Baseline CNN model		85.4%
Tang et al. [10]		CNN with MFCC features		90.2%
de Andrade et al. [12]		CNN with 80-band mel spectrograms		94.1%
Zhang et al. [11]		Depthwise separable CNN		95.4%
Choi et al. [15]		CNN with temporal convolution		96.1%
Zeng et al. [13]		DenseNet with BiLSTM layers		96.2%
Lin et al. [14]		EdgeSpeechNets		96.3%
Mittermaier et al. [16]		Parametrized sinc-convolutions		96.4%
Majumdar et al. [17]		MatchboxNet		97.5%
Li et al. [18]		Wide-ResNet with MoEx		98.0%
Ours (Siamese)		CNN with dropout, 40-band mel spectrograms		94.5%
Ours (Siamese), 8 kHz		Same as above, downsampled recordings		92.0%

**Table 4 sensors-21-08313-t004:** Keyword detection performance for models trained and evaluated in monolingual scenarios. The Area Under Precision–Recall Curve is based on the keyword classes present in the search patterns. We report it either with micro- or macro-averaging. The F-score column represents the best result achieved by the model across different settings of the detection threshold. All models use the default detector configuration.

**Model**	**Training ^1 _(# KW)_^**	**Patterns ^2^**	**Evaluation ^3 _(# KW)_^**	**AUPRC**		**F-Score**
				* **Micro** *	* **Macro** *	
**English → English**						
Siamese	SC 1_EN^(30)^_	SC 1_EN_	SC 1_EN,mix^(30)^_	91.3%	91.3%	0.94
Siamese	SC 2_EN_, CV_EN^(37)^_	SC 1_EN_	SC 1_EN,mix^(30)^_	87.9%	88.0%	0.92
Siamese	SC 1_EN_, SWC_EN^(4K+)^_	SC 1_EN_	SC 1_EN,mix^(30)^_	69.4%	65.6%	0.79
Siamese	ALL_EN^(4K+)^_	SC 1_EN_	SC 1_EN,mix^(30)^_	70.8%	67.4%	0.81
Prototypical	SC 1_EN^(30)^_	SC 1_EN_	SC 1_EN,mix^(30)^_	80.2%	81.5%	0.88
Siamese	SC 1_EN^(30)^_	SC 1_EN_	SC 1_EN,Vox^(30)^_	60.4%	65.0%	0.62
Siamese	SC 2_EN_, CV_EN^(37)^_	SC 1_EN_	SC 1_EN,Vox^(30)^_	55.2%	62.4%	0.57
Siamese	SC 1_EN_, SWC_EN^(4K+)^_	SC 1_EN_	SC 1_EN,Vox^(30)^_	38.0%	43.4%	0.47
Siamese	ALL_EN^(4K+)^_	SC 1_EN_	SC 1_EN,Vox^(30)^_	43.7%	46.9%	0.50
Prototypical	SC 1_EN^(30)^_	SC 1_EN_	SC 1_EN,Vox^(30)^_	40.8%	50.2%	0.54
Siamese	SC 1_EN^(30)^_	ΔSC_EN_	ΔSC_EN,mix^(5)^_	48.3%	53.1%	0.65
Prototypical	SC 1_EN^(30)^_	ΔSC_EN_	ΔSC_EN,mix^(5)^_	39.6%	42.3%	0.59
**Polish → Polish**						
Siamese	ALL_PL^(5K+)^_	KW_PL_	KW_PL,mix^(22)^_	8.8%	1.3%	0.20
Prototypical	ALL_PL^(5K+)^_	KW_PL_	KW_PL,mix^(22)^_	5.9%	1.9%	0.22
Siamese	ALL_PL^(5K+)^_	KW_PL_	KW_PL,real^(22)^_	0.0%	0.0%	0.02
Prototypical	ALL_PL^(5K+)^_	KW_PL_	KW_PL,real^(22)^_	0.0%	0.0%	0.01

1: Training data consists of utterances from Speech Commands (SC 1_EN_, SC 2_EN_), Common Voice (CV_EN_), and Spoken Wikipedia Corpus (SWC_EN_). We also use a combined dataset (ALL_EN_). For the Polish language, we use all accessible training data (ALL_PL_), i.e.,: CV_PL_, WUT_PL_ and TTS_PL_. Number of keyword classes (*#* KW) is denoted with a subscript. 2: English search patterns are extracted from Speech Commands training data (SC 1_EN_) or from the subset present only in the second version of the dataset (ΔSC_EN_). Polish templates come from the target keywords dataset (KW_PL_). We use 10 examples for each keyword class. 3: English models are evaluated with 30 keyword classes, either on fully synthetic mixtures of Speech Commands utterances (SC 1_EN,mix_) or Speech Commands keywords overlaid on VoxCeleb recordings (SC 1_EN,Vox_). We also show the performance on the “*delta*” dataset mixtures (ΔSC_EN,mix_, i.e., 5 classes). Polish evaluations assess 22 keywords either in synthetic mixtures (KW_PL,mix_) or actual YouTube audio streams (KW_PL,real_).

**Table 5 sensors-21-08313-t005:** Keyword detection performance for models trained and evaluated in cross-lingual scenarios. Metric values are reported in the same manner as in Table 4.

**Model**	**Training ^1 _(# KW)_^**	**Patterns ^2^**	**Evaluation ^3 _(# KW)_^**	**AUPRC**		**F-Score**
				* **Micro** *	* **Macro** *	
**English → Polish**						
Siamese	SC 1_EN^(30)^_	KW_PL_	KW_PL,mix^(22)^_	20.3%	10.6%	0.39
Siamese	SC 2_EN_, CV_EN^(37)^_	KW_PL_	KW_PL,mix^(22)^_	10.3%	8.9%	0.31
Siamese	SC 1_EN_, SWC_EN^(4K+)^_	KW_PL_	KW_PL,mix^(22)^_	15.5%	11.6%	0.34
Siamese	ALL_EN^(4K+)^_	KW_PL_	KW_PL,mix^(22)^_	7.3%	10.7%	0.26
Prototypical	SC 1_EN^(30)^_	KW_PL_	KW_PL,mix^(22)^_	13.4%	2.6%	0.27
Siamese	SC 1_EN^(30)^_	KW_PL_	KW_PL,real^(22)^_	0.2%	0.2%	0.03
Siamese	SC 2_EN_, CV_EN^(37)^_	KW_PL_	KW_PL,real^(22)^_	0.1%	0.2%	0.02
Siamese	SC 1_EN_, SWC_EN^(4K+)^_	KW_PL_	KW_PL,real^(22)^_	0.1%	0.1%	0.02
Siamese	ALL_EN^(4K+)^_	KW_PL_	KW_PL,real^(22)^_	0.1%	0.1%	0.01
Prototypical	SC 1_EN^(30)^_	KW_PL_	KW_PL,real(22)_	0.3%	0.2%	0.04
**Combined (English + Polish) → Polish**						
Siamese	ALL_EN_, ALL_PL^(9K+)^_	KW_PL_	KW_PL,mix^(22)^_	6.5%	7.2%	0.23
Prototypical	ALL_EN_, ALL_PL^(9K+)^_	KW_PL_	KW_PL,mix^(22)^_	7.6%	2.7%	0.22
Siamese	ALL_EN_, ALL_PL^(9K+)^_	KW_PL_	KW_PL,real^(22)^_	0.1%	0.1%	0.02
Prototypical	ALL_EN_, ALL_PL^(9K+)^_	KW_PL_	KW_PL,real^(22)^_	0.2%	0.0%	0.04

1: Training data are denoted in the same way as in Table 4. 2: All search patterns come from the Polish target keywords dataset (KW_PL_). 3: Evaluation is performed on Polish mixtures (KW_PL,mix_) and real recordings (KW_PL,real_) with 22 keywords.

**Table 6 sensors-21-08313-t006:** Keyword detection performance for the pre-trained speech embedder. We use it either with default detection settings or with an additional post-processing procedure. We also compare the embeddings generated directly from the pre-trained model and from a model fine-tuned on selected datasets. Metric values are reported in the same manner as in Table 4.

**Model**	**Training ^1 _(# KW)_^**	**Patterns ^2^**	**Evaluation ^3 _(# KW)_^**	**AUPRC**		**F-Score**
				* **Micro** *	* **Macro** *	
**Speech embedder (English), pre-trained**						
Google	—	SC 1_EN_	SC 1_EN,mix^(30)^_	15.3%	71.5%	0.41
Google	—	SC 1_EN_	SC 1_EN,Vox^(30)^_	3.7%	38.5%	0.07
Google	—	ΔSC_EN_	ΔSC_EN,mix^(5)^_	4.6%	4.3%	0.65
Google	—	KW_PL_	KW_PL,mix^(22)^_	27.2%	62.5%	0.41
Google	—	KW_PL_	KW_PL,real^(22)^_	3.4%	21.0%	0.05
**Speech embedder (English), pre-trained, with post-processing**						
Google	—	SC 1_EN_	SC 1_EN,mix^(30)^_	84.8%	87.7%	0.84
Google	—	SC 1_EN_	SC 1_EN,Vox^(30)^_	40.6%	49.8%	0.48
Google	—	ΔSC_EN_	ΔSC_EN,mix^(5)^_	86.5%	87.8%	0.85
Google	—	KW_PL_	KW_PL,mix^(22)^_	46.2%	53.9%	0.69
Google	—	KW_PL_	KW_PL,real^(22)^_	6.6%	9.2%	0.17
**Speech embedder (English), fine-tuning**						
Google	SC 1_EN^(30)^_	SC 1_EN_	SC 1_EN,mix^(30)^_	15.0%	30.1%	0.30
Google	SC 1_EN^(30)^_	SC 1_EN_	SC 1_EN,Vox^(30)^_	1.2%	1.8%	0.03
Google	WUT_PL^(36)^_	KW_PL_	KW_PL,mix^(22)^_	18.4%	19.9%	0.36
Google	WUT_PL^(36)^_	KW_PL_	KW_PL,real^(22)^_	0.7%	2.8%	0.02
**Speech embedder (English), fine-tuning, with post-processing**						
Google	SC 1_EN^(30)^_	SC 1_EN_	SC 1_EN,mix^(30)^_	24.4%	25.7%	0.42
Google	SC 1_EN^(30)^_	SC 1_EN_	SC 1_EN,Vox^(30)^_	1.3%	1.8%	0.04
Google	WUT_PL^(36)^_	KW_PL_	KW_PL,mix^(22)^_	15.5%	16.0%	0.37
Google	WUT_PL^(36)^_	KW_PL_	KW_PL,real^(22)^_	1.3%	1.5%	0.05

1: The speech embedding model [6] is pre-trained on English YouTube audio clips. In most experiments, we use the generated embeddings directly, without any further training of the model. In fine-tuning experiments, we use utterances from the Speech Commands dataset (SC 1_EN_) or our own recordings (WUT_PL_). 2,3: Search patterns and evaluation recordings are denoted in the same way as in Table 4.

**Table 7 sensors-21-08313-t007:** Keyword detection performance on the final evaluation dataset (CC_PL,real_).

**Model (Detector)**	* **Micro** *			* **Macro** *		
	**Precision**	**Recall**	**F-Score**	**Precision**	**Recall**	**F-Score**
Google	0.4%	18.5%	0.01	14.2%	25.6%	0.18
Google (post)	3.6%	17.0%	0.06	8.2%	22.2%	0.12

## Data Availability

Publicly available datasets: Speech Commands v1: http://download.tensorflow.org/data/speech_commands_v0.01.tar.gz (accessed on 10 December 2021); Speech Commands v2: http://download.tensorflow.org/data/speech_commands_v0.02.tar.gz (accessed on 10 December 2021); Mozilla Common Voice: https://commonvoice.mozilla.org/en/datasets (accessed on 10 December 2021); Spoken Wikipedia Corpus: https://nats.gitlab.io/swc (accessed on 10 December 2021); The YouTube fragments with Polish keywords are available on request for research purposes.

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
