# Peer review of "Generalisation Gap of Keyword Spotters in a Cross-Speaker Low-Resource Scenario"

_sensors, 2021, doi:10.3390/s21248313_

Round 1
Reviewer 1 Report
In this manuscript, the authors describe the details of development of a Polish keyword spotting system for call centre recordings. Given the limited data available for Polish recordings, the authors use acoustic similarity matching based approaches to build the systems. More specifically, Siamese and prototypical neural networks are used to train the models. While the authors observe usable results in English, the model performance on Polish speech is poor. The authors further use a generic pre-trained model provided by Google to extract audio embeddings. Although the results on Polish synthetic recordings are promising, the performance on realistic scenarios is still poor. Some further analyses have been done to look at the visualization of the generated embeddings.
The authors have done a detailed investigation on the Polish keyword spotting task with limited data, including a relatively complete review of previous work, comparison between different modeling approaches and data sets, and analysis of learned representation. However, in terms of novelty, model descriptions and experimental results, there are still several gaps, which need to be carefully addressed.
Comments:
- Novelty:
Not many novel approaches have been proposed in the manuscript, and instead, many existing approaches have been tried for the Polish keyword spotting tasks. Therefore, the reviewer would suggest the authors add more novel methods to alleviate the limited data issue, and clearly state your novelty in the manuscript.
- Model description:
In section 2.2, the authors introduce the Siamese network and Prototypical network used for model training. Could you add more details on their loss functions, e.g. mathematical definition?
- Experimental results:
The results are shown for both “Mix” and “Real” Polish test sets. It’s not clear for the reviewer to understand why the results on “Mix” and “Real” data sets are very different. Could the authors provide more detailed analysis (i.e. key factor) on that?
All the results in this manuscript show relatively poor performance on Polish test sets. Therefore, It would be much more convincing if the authors can add more experiments, which can provide some positive results indicating clear improvement of the Polish keyword spotting system.
- Other minor issues:
Since the authors provide a detailed review of previous research on keyword spotting, could you also add the following paper as a reference:
Guo J, Kumatani K, Sun M, et al. Time-delayed bottleneck highway networks using a dft feature for keyword spotting[C]//2018 IEEE International Conference on Acoustics, Speech and Signal Processing (ICASSP). IEEE, 2018: 5489-5493.
Reviewer 2 Report
The idea and methodology are probably interesting and correct for researchers workin in this field.
In terms of presentation and background theory the paper can be more concise and accurate.
Reviewer 3 Report
This is a paper on an interesting topic: Keyword spotting in low-resource languages, here Polish. The authors attempt various few-shot approaches to solve this problem, also exploiting larger English-language corpora for transfer learning. They find that none of their methods produce good results, with the pure few-shot approaches being largely unusable, and the best model being based on Google embeddings trained on larger English-language corpora.
The paper is well-written, but somewhat inconsequential. There are many mentions being made of influence factors that are not scientific in nature, e.g. lacking data availability, promising approaches being released or spotted too late, or lack of time to expand the research further. Unfortunately, this shows in the results. The paper mainly presents negative results. I do not believe that this is necessarily a bad thing, but the results are not sufficiently analyzed and do not serve as a basis for future improvements. Even in places where keyword clusters are visualized in latent space, the authors do not find any real conclusions. As such, I am not sure what purpose this paper serves to the research community.
This is particularly frustrating because there are many factors at play here that could, in fact, serves as interesting inspiration for future research work if analyzed properly. First and foremost, I suspect that channel conditions are a major cause of the unsatisfying outcome. This is apparent with the 8kHz vs. 16kHz discussion that is glossed over in section 2, or in the mention of YouTube audio quality in section 3.3. Models adapt to the training condition, and will then be confused by a train-test distribution mismatch easily. Further proof of this lies in the higher robustness of the Google model, as this is trained on a larger variety data. A relatively simple solution for this would lie in the diversification of training data, or data augmentation if that is not possible.
Second, the currently chosen representation does not account for time-stretching and, particularly, pitch-shifted versions of the keyword productions. A deep model may be able to learn pitch-independent representations by itself, but especially for the low-resource case, this could be alleviated by looking towards existing methods (e.g. cepstrum-based representations).
Also, the differences between the Siamese and the prototypical approach are not discussed. It is also not mentioned on what range of keywords the models are trained - with a low number of keywords, they may overfit, which would explain the bad performance on unseen keywords.
Finally, keyword length will heavily contribute to the results as well, as the authors noted themselves at the end with regard to longer phrases. Short keywords (<~5 phonemes) are notoriously hard to spot reliably, and it is worth considering if it is even necessary to include them instead of adding on heuristics to remove false positives.
Some detail observations and questions:
- section 1.2.1: Another drawback of LVCSR is, of course, that wrong transcriptions cannot be recovered later (e.g. by lowering a detection threshold)
- I am no expert on Polish-language training data, but there are definitely other data sets around (e.g. from LDC)
- How is the frame size motivated? This seems dependent on keyword length
- End of section 1.2.2, searching for keywords based on other factors than pure phonetic content: This is an interesting thought and may warrant further consideration. In this particular research work, however, this does not seem to be desired.
- Section 1.3, argument that pattern-matching keyword spotting should be largely language-independent: I would argue against that, e.g. considering the different phonetic inventory of languages as well as tonal languages
- Section 1.4.1: I think this is too broad; SOTA sections on keyword spotting and on low-resource ASR (+ ASR for Polish) would be sufficient
- Section 1.4.2: Why not adapt one of the successful keyword spotting approaches to Polish directly, e.g. via transfer learning/domain adaptation?
- What about exploiting datasets of structurally and phonetically similar languages to Polish (other Slavic languages, maybe especially Russian)?
- How would the approach scale, in theory, for a large set of keywords?
- Section 2.1: Provide list of keywords and why they were chosen
- Section 2.2: Provide model schematics; is the protoypical approach comparable to the Siamese one if the training configuration is so different?
- Section 2.3: What keyword templates are audio frames compared to? Just one keyword production or multiple ones?
- Why are the few-shot models not used directly for the detection task?
- The comparison between the few-shot models and the Google embeddings are somewhat "unfair" because the first two have been optimized for a comparison task (e.g. pushing similar words further apart rather than moving them together in latent space); even more surprising that the embedding approach outperforms them
- Section 2.3.2: What are the "distinct characteristics" of the Google model?
- Section 3: What do the confusions show? Are they phonetically similar or completely random?
- Section 3.3: What about finetuning English models on Polish in a few-shot manner (see e.g. https://arxiv.org/abs/2008.11228 for an NLP example, there are probabl also ASR approaches around)
- Section 3.4: How was the finetuning implemented?
Reviewer 4 Report
The authors presented the development of a keyword spotting system detecting regions of interest in Polish call centre recordings. The paper is interesting, and it is related to an important field. However, the language must be proofread by a native speaker. Some major comments are below:
- The abstract must be improved with a stating background sentence;
- The introduction is not clearly presenting the purpose and scope of the study;
- The material and methods section needs to define the statistical analysis to be performed for the presentation of the results;
- The results are interesting and compared;
- The discussion must be improved, and a brief conclusions' section must be added.
Reviewer 5 Report
The article is reasonably well written. Usable as introductory material. Fairly interesting subject (albeit for a limited audience).
WOuld be nice to see results on SNIPS and the Mobvoi Hotwords (http://www.openslr.org/87/), albeit I'm not sure about the latter if it's English and not Mandarin.
I'm rating low on significance of content because of the limited audience.
Round 2
Reviewer 1 Report
The authors have carefully replied to the reviewer’s comments. As the authors point out, this paper is more like a proof-of-concept work to investigate different methods for Polish keyword spotting tasks in low-resource conditions. The main contribution is to conduct practical evaluation of existing approaches on these tasks, and compare the results. The authors also add more details on the selected loss functions and provide precise descriptions. The experimental design is also better described in the revised manuscript.
There are still a few minor issues that need to be addressed:
- The reviewer thinks that it would be better to clearly state your novelty at the beginning of your manuscript. Section 1.2 already shows the paper overview, but more specific contributions should be clearly stated. Since this manuscript hasn’t proposed any new techniques, other contributions (e.g. dataset creation and evaluation, comparison between existing approaches, first experiments on Polish data, etc) should be pointed out.
- The reviewer would suggest adding some novel approaches or solutions in the future work section. These novel approaches could be based on solving the existing issues shown in the manuscript.
Author Response
Thank you for your appreciation of our changes.
Our answer for the remaining issues:
(1) The reviewer thinks that it would be better to clearly state your novelty at the beginning of your manuscript. Section 1.2 already shows the paper overview, but more specific contributions should be clearly stated. Since this manuscript hasn’t proposed any new techniques, other contributions (e.g. dataset creation and evaluation, comparison between existing approaches, first experiments on Polish data, etc) should be pointed out.
Answer:
Thank you, we agree on this matter. As suggested, we have highlighted our contributions in section 1.3. We think our work can be summarised in the following contributions:
- We create a number of experimental protocols (mono- and cross-lingual) for keyword spotting in continuous audio recordings.
- We prepare additional Polish datasets for training and evaluation purposes. The recordings sourced from YouTube clips are available on request.
- We evaluate similarity ranking models (Siamese and prototypical networks) in practical keyword detection tasks.
- We report the results on using the Google speech embedder on Polish data.
- We compare the embedding spaces generated by different combinations of the models and training data.
- We highlight the gap between popular keyword classification benchmarks and performance in more practical tasks (analysing authentic call centre recordings).
(2) The reviewer would suggest adding some novel approaches or solutions in the future work section. These novel approaches could be based on solving the existing issues shown in the manuscript.
Answer:
We have reorganised chapter 4 (discussion) to separate summary and future work subsections. In the future work section we have expanded our descriptions and provided additional approaches that we think could be useful in solving the issues we have found in our experiments, such as active learning approaches improving the efficiency of data annotation.
We've attached the latexdiff file, where all our changes are highlighted.

Reviewer 3 Report
I appreciate the changes taken to clarify certain key points about the research work, and present a somewhat more in-depth analysis. Nevertheless, I do not think that this changes the fundamental flaw of the paper, i.e. presenting a negative result that does not really serve as a basis for future improvements because the reasons are manifold and could not be analyzed in enough detail. In particular, I mean the flaws that came about because certain data was not available to the researchers or they simply did not know about it, and more approaches that were not tested because time ran out. To round the paper out, I think many more experiments and in-depth analyses would be necessary to make generalizable statements. In other words, I imagine building a functioning system for Polish would be trivial with enough data. If this is not doable, the focus should be on remedying this data scarcity.
Author Response
Thank you for your comment. We respect and understand the presented point of view. We also vote for publishing negative results, and we think such articles are valuable to society, if only to avoid repeating similar unfruitful attempts. However, we think that our contribution is a bit broader than that, as it lies in creating and labelling new datasets (available on request), checking different existing approaches on a practical problem and proposing numerous adjustments, for the first time evaluating popular off-the-shelf embedding solution in a cross-lingual setting. It also highlights the issue that searching for keywords in an audio stream is much more challenging to solve using algorithms based on neural networks than classifying separated words. We have added such comments in the Discussion section.
We've attached the latexdiff file, where all our changes are highlighted.

Reviewer 4 Report
The authors considered the previous comments, and the manuscript can be accepted.
Author Response
Thank you for your appreciation of our work.
We've attached the latexdiff file, where all our changes are highlighted.
